# Wide reference databases for typing *Trypanosoma cruzi* based on amplicon sequencing of the minicircle hypervariable region

**Fanny Rusman[1]ᵒ, Anahí G. Díaz[1]ᵒ, Tatiana Ponce[1], Noelia Floridia-Yapur[1], Christian Barnabé[2], Patricio Diosque[1]\*, Nicolás Tomasini[1]\***

**1** Unidad de Epidemiología Molecular (UEM), Instituto de Patología Experimental Dr. Miguel Ángel Basombrío, Universidad Nacional de Salta-CONICET, Salta, Salta, Argentina, **2** Institut de Recherche pour le Développement (IRD), UMR INTERTRYP IRD-CIRAD, University of Montpellier, Montpellier, France

ᵒ These authors contributed equally to this work.
\* patricio.diosque@unsa.edu.ar (PD); nicolas.tomasini@conicet.gov.ar (NT)

## Abstract

### Background

*Trypanosoma cruzi*, the etiological agent of Chagas Disease, exhibits remarkable genetic diversity and is classified into different Discrete Typing Units (DTUs). Strain typing techniques are crucial for studying *T. cruzi*, because their DTUs have significant biological differences from one another. However, there is currently no methodological strategy for the direct typing of biological materials that has sufficient sensitivity, specificity, and reproducibility. The high diversity and copy number of the minicircle hypervariable regions (mHVRs) makes it a viable target for typing.

### Methodology/Principal findings

Approximately 24 million reads obtained by amplicon sequencing of the mHVR were analyzed for 62 strains belonging to the six main *T. cruzi* DTUs. To build reference databases of mHVR diversity for each DTU and to evaluate this target as a typing tool. Strains of the same DTU shared more mHVR clusters than strains of different DTUs, and clustered together. Different identity thresholds were used to build the reference sets of the mHVR sequences (85% and 95%, respectively). The 95% set had a higher specificity and was more suited for detecting co-infections, whereas the 85% set was excellent for identifying the primary DTU of a sample. The workflow's capacity for typing samples obtained from cultures, a set of whole-genome data, under various simulated PCR settings, in the presence of co-infecting lineages and for blood samples was also assessed.

### Conclusions/Significance

We present reference databases of mHVR sequences and an optimized typing workflow for *T. cruzi* including a simple online tool for deep amplicon sequencing analysis

**Data Availability Statement:** The data are available for download at the Sequence Read Archive (SRA)

database under the accession number
PRJNA514922.

**Funding:** The current study is funded by the
National Scientific and Technical Research Council
(CONICET, Argentina), Award number: PUE-2016
to PD, and the National Agency for Scientific and
Technological Promotion (ANPCyT), award
number: PICT2019-02855 to NT. The funders had
no role in study design, data collection and
analysis, decision to publish, or preparation of the
manuscript.

**Competing interests:** The authors have declared
that no competing interests exist.

(https://ntomasini.github.io/cruzityping/). The results show that the workflow displays an equivalent resolution to that of the other typing methods. Owing to its specificity, sensitivity, relatively low cost, and simplicity, the proposed workflow could be an alternative for screening different types of samples.

## Author summary

Chagas disease, caused by the parasite *Trypanosoma cruzi*, is a significant public health concern in Latin America. This parasite is genetically diverse and classified into different lineages. Proper strain typing techniques are necessary to study *T. cruzi*, because their lineages have significant biological differences. Several typing methods have been proposed, each of which has its own strengths and limitations. However, most of these methods lack sensitivity or fail for discriminating some lineages. Genetic markers with high copy numbers are required to gain sensitivity. Here, we deep sequenced DNA regions present in the large mitochondrion of the parasite (mHVRs) from strains belonging to the six main lineages to obtain reference mHVR sequences and develop a typing workflow. Amplicon sequencing of mHVR was conducted on 62 *T. cruzi* strains. Despite high sequence diversity, strains of the same lineage shared more sequences than strains of different lineages. Two reference sets of mHVR sequences were generated and evaluated for their ability to typify distinct types of *T. cruzi* samples. The workflow presented in this study could serve as a valuable resource for *T. cruzi* typing in future studies.

## Introduction

*Trypanosoma cruzi*, a flagellate parasite belonging to the class Kinetoplastea and the Trypanosomatidae family, is the etiological agent of Chagas Disease. This neglected tropical disease affects around 6 to 7 million individuals worldwide, predominantly in Latin America [1].

*T. cruzi* exhibits remarkable genetic diversity, at least six main lineages or Discrete Typing Units (DTUs), named TcI to TcVI, have been recognized according to the current consensus [2,3]. However, in recent years, the seventh lineage associated with bat infections (Tcbat) and closely related to TcI has been proposed [4,5].

Various molecular techniques have been developed to study the genetic diversity of *T. cruzi*. Multilocus Enzyme Electrophoresis (MLEE) was one of the first non-DNA methods used [6]. Later, several DNA typing techniques were developed, including Low Stringency Single Specific Primer (LSSP-PCR) [7]), mini-exon [8], amplification of a single polymorphic locus [9], Multilocus Microsatellites typing (MLMT) [10,11], Restriction Fragment Length Polymorphism (RFLP-PCR) [12], PCR schemes [13–15], and Multilocus Sequence Typing (MLST) [16–19]. Some recently developed typing approaches have shown promising results, such as deep amplicon sequencing of mini exon genes or minicircle hypervariable regions, and genome-wide locus sequence typing (GLST) [20–23].

Due to the low level of parasites circulating in the peripheral blood or infected tissues in chronically infected patients, most typing methods have limited sensitivity [24,25]. At this point, genetic markers with a high number of copies are required to achieve adequate detection sensitivity. Like other kinetoplastids, *T. cruzi* has a single mitochondrion with a unique mitochondrial DNA called kinetoplast (kDNA) [26]. The kDNA network consists of two types of topologically interlocked DNA circles: maxicircles (≈20-40kb) and minicircles (≈1.4kb).

Per network, there are around $2x10^4$ minicircles, which represents approximately 20–25% of the whole cellular DNA [27,28]. Minicircle sequences consist of four highly conserved regions (mHCRs) of ≈120bp intercalated by an equal number of hypervariable regions (mHVRs) of ≈240bp [29,30]. mHVRs have been extensively used as PCR targets for *T. cruzi* DNA detection with good sensitivity and specificity [31]. The amplicons were amplified by primers annealing the mHCRs flanking the mHVR, so the entire mHVR sequence remained included in the amplicon [32]. There is robust evidence that a set of mHVR sequences is lineage and genotype-specific (at the intra-lineage level [33,34]. In a previous study, based on deep sequencing of the minicircle hypervariable regions of kDNA, we suggested a strategy for typing and elucidating the intra-specific diversity of *T. cruzi*. The diversity of mHVR sequences in nine reference strains from the six major DTUs was preliminarily evaluated and compared to establish such a typing approach. A large number of *T. cruzi* strains can be typed simultaneously using the mHVR-amplicon sequencing method among the advantages of this technique [21]. In the present work, we broadened the application of the previous amplicon sequencing approach, presenting an optimized typing workflow based on deep sequencing of mHVR amplicons from a wide panel of 62 strains belonging to the six main DTUs. We additionally provide reference databases of mHVR sequences and a simple tool for bioinformatic analysis. Finally, PCR and sequencing protocol and the bioinformatic steps were evaluated in clinical samples.

## Materials and methods

### Strains and blood samples

DNA from 52 *T. cruzi* strains belonging to the six main DTUs was analyzed in this study (Table 1). Sequences of ten additional strains previously analyzed by Rusman et al., [21] were also included. Twenty-eight blood samples were obtained from a previous cross-sectional study conducted in February 2010 in El Palmar (27˚ 40 32,700S; 61˚ 340 19,900W), a settlement located in the 12 de Octubre Department, Chaco Province (Argentina) [34]. The protocol was approved by the Bioethics Committee of the Faculty of Health Sciences at the National University of Salta, Argentina. Blood was preserved in guanidine-EDTA buffer (Five milliliters of blood mixed with an equal volume of a solution of 6 M-HCl and 0.2 M EDTA). A standard phenol-chloroform method was used for DNA extraction.

### mHVR sequencing

The minicircle hypervariable regions of the strains were amplified as described by Rusman et al., [21]. To generate mHVR libraries from the blood samples, two consecutive PCR reactions were performed, each with a volume of 15μl. The first reaction mixture included 200nM of modified primers 121 and 122 described by Rusman et al., [21], 3μl of DNA, 0.375U of Fast Start High Fidelity Enzyme Blend (Roche), 1X buffer supplied with the enzyme blend, 4.5mM of MgCl2 (Roche), 5% DMSO (Roche), and 0.2mM of PCR grade nucleotide mix (Roche). This PCR protocol started with an initial denaturation for 3 min at 94˚C, followed by two cycles of 97.5˚C for 1 min and 64˚C for 2 min. Then, 33 cycles of 94˚C for 1 min and 64˚C for 1 min were run, with a final extension at 72˚C for 10 min. For the second reaction, which aimed to incorporate barcodes into the first reaction amplicons, the mixture contained 200nM of each barcode, 2μl of the primary amplicon, 0.375U of Fast Start High Fidelity Enzyme Blend (Roche), 1X buffer supplied with the enzyme blend, 4.5mM of MgCl2 (Roche), 5% DMSO (Roche), and 0.2mM of PCR grade nucleotide mix (Roche). The protocol for this reaction was as follows: initial denaturation for 3 min at 95˚C, followed by eight cycles of 95˚C for 30 s, 55˚C for 30 s, and 72˚C for 30 s, ending with a final extension at 72˚C for 5 min.

**Table 1. Strains used in this work.**

| Strain | DTU | Origin | Host |
|---|---|---|---|
| 1. LL0553R2cl3 | TcI | Argentina | *Triatoma infestans* |
| 2. PalDa20cl3* | TcI | Argentina | *Didelphis albiventris* |
| 3. PalDa30V2cl2 | TcI | Argentina | *Didelphis albiventris* |
| 4. PalDa4 | TcI | Argentina | *Didelphis albiventris* |
| 5. TeDa2cl4* | TcI | Argentina | *Didelphis albiventris* |
| 6. TEV55cl1* | TcI | Argentina | *Triatoma infestans* |
| 7. 86/2021 | TcI | Bolivia | *Coendou prehensilis* |
| 8. P209cl1 | TcI | Bolivia | *Homo sapiens* |
| 9. QRA05 | TcI | Bolivia | *Triatoma infestans* |
| 10. SO40 | TcI | Bolivia | *Triatoma infestans* |
| 11. CUICAcl1 | TcI | Brazil | *Philander opossum* |
| 12. CUTIAcl1 | TcI | Brazil | *Dasyprocta aguti* |
| 13. SilvioX10/7 | TcI | Brazil | *Homo sapiens* |
| 14. SP104cl1 | TcI | Chile | *Triatoma spinolai* |
| 15. Vincho111 | TcI | Chile | *Triatoma infestans* |
| 16. VQUI1 | TcI | Chile | *Triatoma infestans* |
| 17. 393TA | TcI | Colombia | *Rattus rattus* |
| 18. Colombiana | TcI | Colombia | *Homo sapiens* |
| 19. MR-C | TcI | Colombia | *Homo sapiens* |
| 20. NS | TcI | Colombia | *Homo sapiens* |
| 21. ElSalvador1980 | TcI | El Salvador | *Homo sapiens* |
| 22. R143 | TcI | Guyana | *Panstrongylus geniculatus* |
| 23. DAVIS | TcI | Honduras | *Triatoma dimidiata* |
| 24. ARMADILLO1973 | TcI | USA | *Dasypus novemcinctus* |
| 25. DM28c | TcI | Venezuela | *Didelphis marsupialis* |
| 26. Saimiri4a | TcI | Venezuela | *Saimiri sciureus* |
| 27. TU18cl93* | TcII | Bolivia | *Triatoma infestans* |
| 28. Bug2150 | TcII | Brazil | *Triatoma infestans* |
| 29. Bug2152 | TcII | Brazil | *Triatoma infestans* |
| 30. Esmeraldo* | TcII | Brazil | *Homo sapiens* |
| 31. MAS1cl1 | TcII | Brazil | *Homo sapiens* |
| 32. X-300 | TcII | Brazil | *Homo sapiens* |
| 33. CBBcl4 | TcII | Chile | *Homo sapiens* |
| 34. IVVcl4 | TcII | Chile | *Homo sapiens* |
| 35. LL0513R2 | TcIII | Argentina | *Triatoma infestans* |
| 36. LL051P24RI | TcIII | Argentina | *Canis familiaris* |
| 37. M5631cl5 | TcIII | Brazil | *Dasypus novemcinctus* |
| 38. M6241cl6 | TcIII | Brazil | *Homo sapiens* |
| 39. X109/2* | TcIII | Paraguay | *Canis familiaris* |
| 40. CANIIIcl1* | TcIV | Brazil | *Homo sapiens* |
| 41. 92122102R | TcIV | USA | *Procyon lotor* |
| 42. 93053102Rcl3 | TcIV | USA | *Procyon lotor* |
| 43. DogTheis | TcIV | USA | *Canis familiaris* |
| 44. STC10Rcl3 | TcIV | USA | *Procyon lotor* |
| 45. STC13Rcl3 | TcIV | USA | *Procyon lotor* |
| 46. STC16Rcl4 | TcIV | USA | *Procyon lotor* |
| 47. STC5Rcl2 | TcIV | USA | *Procyon lotor* |

*(Continued)*

**Table 1.**  (Continued)

| Strain | DTU | Origin | Host |
|--------|-----|--------|------|
| 48. LL014R1* | TcV | Argentina | *Triatoma infestans* |
| 49. LL0401R0cl1 | TcV | Argentina | *Triatoma infestans* |
| 50. SC43cl1 | TcV | Bolivia | *Triatoma infestans* |
| 51. MIz02 | TcV | Bolivia | *Triatoma infestans* |
| 52. CHUL23 | TcV | Bolivia | *Triatoma infestans* |
| 53. Bug2145 | TcV | Brazil | *Triatoma infestans* |
| 54. MNcl2* | TcV | Chile | *Homo sapiens* |
| 55. SAXP19 | TcV | Peru | *Homo sapiens* |
| 56. LL015P68R0cl4* | TcVI | Argentina | *Canis familiaris* |
| 57. TeP6 | TcVI | Argentina | *Canis familiaris* |
| 58. TeV67 | TcVI | Argentina | *Triatoma infestans* |
| 59. VM09 | TcVI | Bolivia | *Triatoma infestans* |
| 60. CL Brener | TcVI | Brazil | *Triatoma infestans* |
| 61. Tulacl92 | TcVI | Chile | *Homo sapiens* |
| 62. P63cl1 | TcVI | Paraguay | *Triatoma infestans* |

* Reads obtained from a previous study [21].

The Agentcourt AMPure XP-PCR Purification kit (Beckman Genomics, USA) was then used to purify the amplicons. Qubit Fluorometer 2.0 (Invitrogen, USA) was used to measure the concentration of the purified amplicons. A 5200 Fragment Analyzer System (Advanced Analytical Technologies Inc.- Agilent, USA) was used to validate the estimated size of the libraries as the average size of the mHVR amplicons was ~480bp. The mHVR amplicons from strains were sequenced on an Illumina MiSeq platform and those from blood samples were sequenced on an Illumina NovaSeq platform both using a 500 cycle v2 kit (Illumina, San Diego, USA) at a depth of 80,000 reads per strain. Reads from ten additional samples were obtained from a previous study [21].

## Building the reference datasets

The raw reads were pre-processed, trimmed, P-E merged, and filtered as described in detail by [21]. Then, sequences were clustered at different pairwise identity percentages ranging from 85% to 97.5% every 2.5% increment. This was made by using *"pick_de_novo_otus.py"* script from QIIME v1.9.1 [35]. The parameters were used by default to cluster the sequences according to the two identity thresholds. The outputs (seqs_otus.txt and the otu table) were filtered using *"filter_otus_from_otu_table.py"* script from QIIME v1.9.1 to discard those mHVR clusters with low abundance and conserving those that were observed more than five times. The datasets were first evaluated based on their ability to cluster strains of the same DTU. The most abundant sequence in each mHVR cluster was selected as the representative sequence using the *"pick_rep_set.py"* script from QIIME v1.9.1, with the other parameters by default. The output is a FASTA file containing one representative sequence for each mHVR cluster with their corresponding cluster identifier.

## Using the reference datasets

The reference sets can be used for typing unknown samples. A DTU-tag was assigned to each representative sequence in the reference set according to the DTU in which the mHVR cluster

was observed. If an mHVR cluster is shared by strains of different DTUs, the tag is assigned based on the DTU of the strain, with more reads for that specific mHVR cluster.

Following the mHVR sequencing of the sample(s) to be typified, the processed reads -according to the aforementioned procedure- and one of the reference sets are used to run the "*pick_closed_reference_otus.py*" algorithm available in QIIME 1.9.1 or a Google Colaboratoy notebook implementing the USearch algorithm [36]. The result is a table of mHVR clusters containing each sample.

DTU assignment to each sample is based on the following rules:

1. For each sample, the number and percentage of reads clustered with the DTU-tagged representative sequences is calculated.

2. The DTU-tag with the most reads in the sample is considered to be the infecting DTU in the sample.

3. Minority DTU-tags in the sample were considered as DTUs infecting the sample if the percentage of reads for such a DTU-tag is higher than a specific cutoff. This cutoff is defined depending on the majority DTU-tag in the sample and was calculated by PCR simulation (see below).

### Reference datasets availability and online typing tool

The two reference datasets of mHVRs, generated at 85% and 95% identity thresholds, are accessible at https://ntomasini.github.io/cruzityping. The methodology outlined in the preceding section was automated through a Google Colaboratory notebook, also available at the aforementioned link. This notebook is configured to accept raw data input, execute the described workflow automatically using reference datasets, and generate various graphical representations. An accompanying tutorial was provided to aid users in navigating this too.

### Evaluation of the 95% reference set for strain typing from whole genome data

To evaluate the 95% reference set raw sequences from different genome-sequencing projects were downloaded from the NCBI SRA database. To evaluate the 95% reference set, raw sequences from different genome-sequencing projects were downloaded from the NCBI SRA database to evaluate a representative genome set of DTUs diversity. Considering the short size of minicircles, only genome projects with no fragment size selection previous sequencing were analyzed. Furthermore, the genomes analyzed were reported as previously typified. The files corresponding to the 29 *T. cruzi* strains of the six main lineages were analyzed. The accession numbers are listed in Table 2.

The reads from the whole-genome sequencing projects were processed and analyzed using the Galaxy platform (https://usegalaxy.org/). Paired-end reads generated by Illumina sequencing underwent quality filtering using Trimmomatic [37] with the following parameters: SLIDINGWINDOW:4:20 LEADING:30 TRAILING:30 MINLEN:40. Sequences generated from other platforms were excluded from trimming. The reads were mapped against the reference set of mHVR at 95% similarity using BWA-MEM v.0.7.17.2 [38] with default parameters. The resulting mapping file in the BAM format was evaluated for coverage using BEDtools [39]. Sequencing reads from the different genomes were mapped to mHVR reference sequences generated at a 95% similarity threshold. The mHVR reference sequences with a coverage of 170 bases mapped to sequencing reads at 10X depth were selected. Two different analyses were performed according to the above condition: A- The percentage of lineage-specific sequences

**Table 2. Strains, DTUs, NCBI SRA accession codes of the analyzed whole genome files.**

|     | Strain | DTU | Access code NCBI-SRA |
| --- | --- | --- | --- |
| 1. | TRYCC1522 | TcI | SRR2057774 |
| 2. | TBM3324 Ecuador | TcI | SRR3676267 |
| 3. | TBM3479B1 Ecuador | TcI | SRR3676269 |
| 4. | H1 Texas | TcI | SRR3676271 |
| 5. | V2 Panama | TcI | SRR3676314 |
| 6. | FcHcl1 Colombia | TcI | SRR3676318 |
| 7. | TMB_2798 (non-cloned)* | TcI? | SRR9643438 |
| 8. | JRcl4 | TcI | SRR547646 |
| 9. | Dm28c | TcI | SRR7592211 |
| 10. | S92a | TcII | SRR6357356 |
| 11. | S44a | TcII | SRR6357357 |
| 12. | S23b | TcII | SRR6357358 |
| 13. | S1162a | TcII | SRR6357359 |
| 14. | S154a | TcII | SRR6357360 |
| 15. | S15 | TcII | SRR6357361 |
| 16. | S11 | TcII | SRR6357362 |
| 17. | Ycl4 | TcII | SRR6357364 |
| 18. | Ycl6 | TcII | SRR11845030 |
| 19. | Berenice | TcII | SRR13321697 |
| 20. | Ikiakarora | TcIII | PRJNA595095 |
| 21. | 231 | TcIII | ERR864236 |
| 22. | M6241cl6 | TcIII | PRJNA169677 |
| 23. | CANIIIcl1 | TcIV | SRR1996499 |
| 24. | SOL | TcV | PRJNA661295 |
| 25. | SC43cl1.1 | TcV | SRR11802127 |
| 26. | 9280cl2 | TcV | SRR1996502 |
| 27. | Cl Brener[1] | TcVI | SRR6357354 |
| 28. | Cl Brener[2] | TcVI | PRJNA661279 |
| 29. | Tulacl2 | TcVI | SRR831221 |

*The sample was reported as non-clonal in the NCBI database, which resulted in TcI in the analyses.

[1] Sequenced with an Illumina HiSeq 2000.

[2] Sequenced with Ion Torrent.

retained in this set of reference sequences was calculated. For instance, if the sequencing reads from a given genome are mapped to 99 mHVR reference sequences from TcI and only one from TcII, this indicates that this genome belongs to the TcI lineage. B- The total number of bases for the reads that mapped to the reference sequences of each lineage. The same analysis was performed for A and B, with a coverage of 270 bases at 10X depth. These procedures were applied to each of the downloaded datasets. A strain level analysis was also made by determining the proportion of mapped 95% reference mHVRs that cluster with each strain in the dataset.

## Analysis of dataset performance on mHVR amplicons

To evaluate the typing resolution of the reference datasets, every strain was typified by using a reference dataset that excluded the strain that was being typified; for example, typing of Sylvio strain is made by a reference set constructed without Sylvio reads. This process was performed

for the 62 strains in this study, and the sensitivity and specificity for typing each DTU were evaluated.

In addition, to evaluate the potential suitability of this workflow for typing biological samples, a PCR simulation algorithm was developed in R (https://github.com/ntomasini/cruzityping/ blob/main/VirtualPCRcode.R) to simulate the stochasticity and efficiency of PCR amplification. The algorithm was based on the basic equation of PCR kinetics proposed by Ruijter et al., [40] but considering the efficiency (e) as a probability of molecule replication instead of a fixed proportion of replicated molecules. First, the algorithm samples $s$ random molecules from a multinomial distribution ($f_0$) according to (1)

$$f_0 = (X_1, \ldots, X_k) \sim M(m_0, p_1, \ldots, p_k) \tag{1}$$

Where $X_k$ is the number of molecules in the mHVR cluster k in the starting DNA of the PCR; $m_0$ is the number of starting molecules in the PCR, and $p_1, \ldots, p_k$ are the probabilities of the mHVR clusters 1 to k defined as the relative frequency of such mHVR clusters in the whole reads for such strain. This step simulates the stochasticity caused by sampling mHVR sequences in the steps before the PCR such as DNA extraction.

Second, the first ten PCR cycles were simulated (the first cycles may introduce bias in mHVR cluster frequencies when few molecules start the reaction and have low efficiency). A binomial distribution is used to simulate the number of molecules that are successfully amplified in each cycle according to $e$ (2).

$$m_i \sim B(n_{i-1}, e) \tag{2}$$

Where $m_i$ is the number of newly synthesized molecules in the $i$-step of the PCR, $n_{i-1}$ is the number of DNA molecules in the previous PCR step, and $e$ is the PCR efficiency defined as a duplication probability for each molecule.

Third, a multinomial distribution is used to determine the identity of the new molecules according to (3)

$$g_i = (X_1, \ldots, X_k) \sim M(m_i, q_1, \ldots, q_k) \tag{3}$$

Where $g_i$ is the set of molecules generated in cycle i, $X_k$ is the number of sequences of cluster $k$ at the end of the PCR cycle, $m_i$ is the number of molecules synthesized in the $i$-cycle, and $q_k$ is the probability of the cluster $k$ defined as the relative frequency of such an mHVR cluster in the strain in the $i$-1 cycle. Finally, the set of newly generated molecules are summed to the previously generated.

$$f_i = f_{i-1} + g_i \tag{4}$$

Where $f_i$ is the resulting set of molecules in cycle $i$ of the PCR. The cycle was iterated until i = 10 and repeated 100 times. Different $m_0$ values (1, 10, and 100 starting DNA molecules) and $e$ (0.7, 0.8, and 0.99) were evaluated. In addition, PCR efficiency is commonly higher than 90% but can be lower in the presence of inhibitors [41], and different values for e (0.7, 0.8, and 0.99) were evaluated to simulate optimal and sub-optimal conditions, which may introduce more stochasticity in cluster abundances. The PCR model was compared to experimental data of mHVR cluster abundances for two independent PCR reactions of the same sample (S1 File).

Because a minority of mHVR clusters were shared among lineages, we used PCR simulation to approximate the probability of false positives for different DTUs and to define cutoffs to reduce such probability to reasonable values. The first ten cycles of a PCR with $m_0 = 100$ and $e = 0.99$ with 100 replicates were simulated. The probability of false positives was calculated for each DTU, as the number of reads clustering to incorrect DTUs was higher than the cutoff.

Different cutoffs were evaluated (0.01–0.05) to reduce, when possible, the error probability of false positives below 0.02.

In addition, mock samples composed of reads of two different strains from different DTUs were evaluated to determine the sensitivity of the reference sets for detecting co-infections. Different proportions of different DTUs were evaluated (95%-5%, 90%-10%, 10%-90%, and 5%-95%). Strains with the highest number of reads were selected to build the mock datasets. The datasets for each strain were sampled according to the expected proportions for each DTU in the mock sample (e.g., 90% of the reads of PalDa20cl3-TcI and 10% of MNcl2-TcV). The mock sample was used as the input in the PCR simulation algorithm using $m_0 = 100$ and $e = 0.99$, with 100 replications. The simulated datasets were typified as described above, and the generated matrix of cutoffs was used to discard false positives. The sensitivity for detecting the less abundant DTU in the sample was evaluated.

## Results

### mHVR clusters are shared among strains within a DTU

To address the suitability of deep amplicon sequencing to genotype *T. cruzi* DTU in a sample, mHVRs of 62 strains from different DTUs were amplified by PCR and deep-sequenced. The number of reads retained after trimming and quality filtering, merging, and more stringent filtering varied between 18,207 and 2,356,494 (S1 Table). The reads were clustered according to sequence similarities using different minimum similarity percentages (85% and 95%) as in a previous work [21]. The number of shared mHVR clusters among different strains is shown in Fig 1. The mHVR clusters are mostly lineage specific. Furthermore, the number of shared clusters among strains decreased when higher similarity thresholds were used. For the 85% similarity threshold, it was observed that TcI strains, which were geographically closer, shared

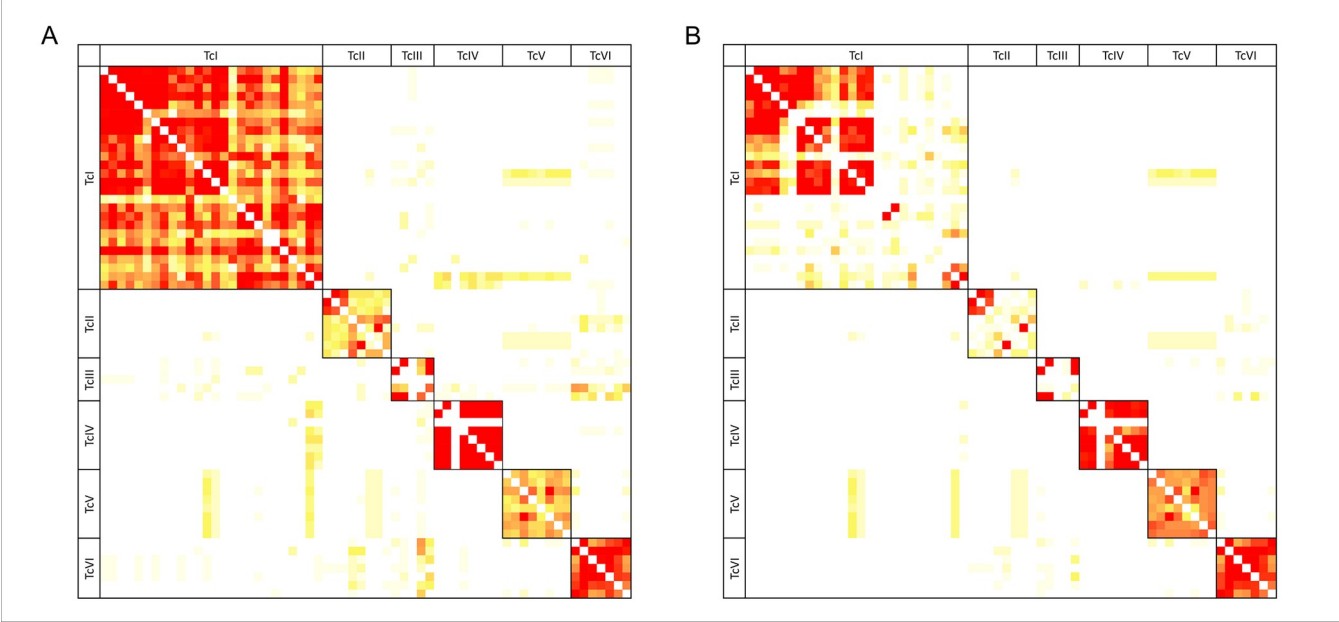

**Fig 1. Strains of the same DTU shared mHVR clusters at different identity thresholds.** Similarity matrices show the number of mHVR clusters shared between strains of the same lineage and between strains of different lineages. Strains were arranged according to their lineage. The color scale indicates the similarity between pairs of strains. A, 85% identity threshold; B, 95% identity threshold. White: 0 shared mHVR clusters, yellow-orange: less than 20 shared mHVR clusters, red: more than 20 shared mHVR clusters.

more mHVR clusters than strains isolated at greater geographical distances. Additionally, at the 95% similarity threshold, most TcI strains shared a few mHVR clusters. In contrast, the TcV and TcVI strains still shared mHVR clusters with other strains of the same DTU. These results suggest that mHVR sequences can be used for typing *T. cruzi* strains.

### Reference sets are suitable for *Trypanosoma cruzi* typing of cultured strains

Six sets of reference sequences were constructed based on the similarity thresholds (85%, 87.5%, 90%, 92.5%, 95% and 97.5%). To evaluate the suitability of the reference sets for typing, each strain was re-typed using a reference set (n– 1) constructed by excluding the sequences of the strain to be typed. Proportions of reads unassigned to any DTU, reads correctly assigned to the DTU of the strain (true positives), and reads erroneously typified to another DTU (false positives) were calculated (S2 Table). In addition, the frequencies of strains that were correctly and incorrectly assigned were calculated (S2 Table). As expected, higher thresholds imply higher specificity in read assignation, although it also implies less sensitivity for DTU assignation to strains (see the 97.5% reference set that failed to assign DTU to five strains in S2 Table). We selected the 95% reference set because it allowed a lower false-positive rate for DTU assignment of reads, in spite of failing to genotype only one TcIII strain. Instead, all n– 1 reference sets constructed with an 85% similarity threshold were able to correctly typify the strains with their corresponding DTU; that is, most of the reads clustered with references of the same DTU. However, the higher rate of false-positive reads in this reference set (S2 Table) may discourage its use in the detection of secondary DTUs in a sample. The proportion of reads clustered for each DTU using 85% and 95% reference sets is shown in Fig 2.

### The 95% reference set is useful for typing strains from whole-genome sequencing data

To address the suitability of the 95% mHVRs reference set for typing, data from different whole-genome sequencing projects were analyzed. The sequences were mapped against the 95% reference set, followed by an evaluation of the mapping coverage and assignment of mHVRs cluster percentages for each lineage. Only sequences with a coverage of at least 170 and 270 bases and a depth greater than or equal to 10X that mapped to reference sequences from each DTU were considered for analysis (Figs 3A and S1A). Also, the percentage of bases that mapped to the reference sequences for each lineage was calculated, excluding regions with a coverage of less than 170 (S1B Fig) and 270 bases and a depth of less than 10X (Figs 3B and S1B). Notably, all evaluated strains were accurately typified using this approach, except for two strains reported as belonging to the TcIII lineage (Ikiakarora and 231). Furthermore, both CL Brener genomes exhibited a high degree of concordance in their typing, despite having been sequenced using different sequencing technologies. In addition, the proportion of mHVR clusters shared between different genomes and strains in the 95% reference dataset was addressed (S2 File). Different patterns were observed within some DTUs for different genomes, suggesting potential utility for intra-DTU typing.

### The suitability of reference sets for typing despite PCR stochasticity

Reference sets of mHVRs are potentially useful for identifying DTUs in biological samples such as blood. However, PCR stochasticity caused by a low amplification efficiency, or a low number of initial DNA molecules may cause the frequency of each mHVR cluster to not represent the real frequency in the sample. To assess the suitability of the reference sets for typing, artificial samples were simulated for each strain in the dataset under different simulated PCR conditions for efficiency and different numbers of initial DNA molecules. The reference sets

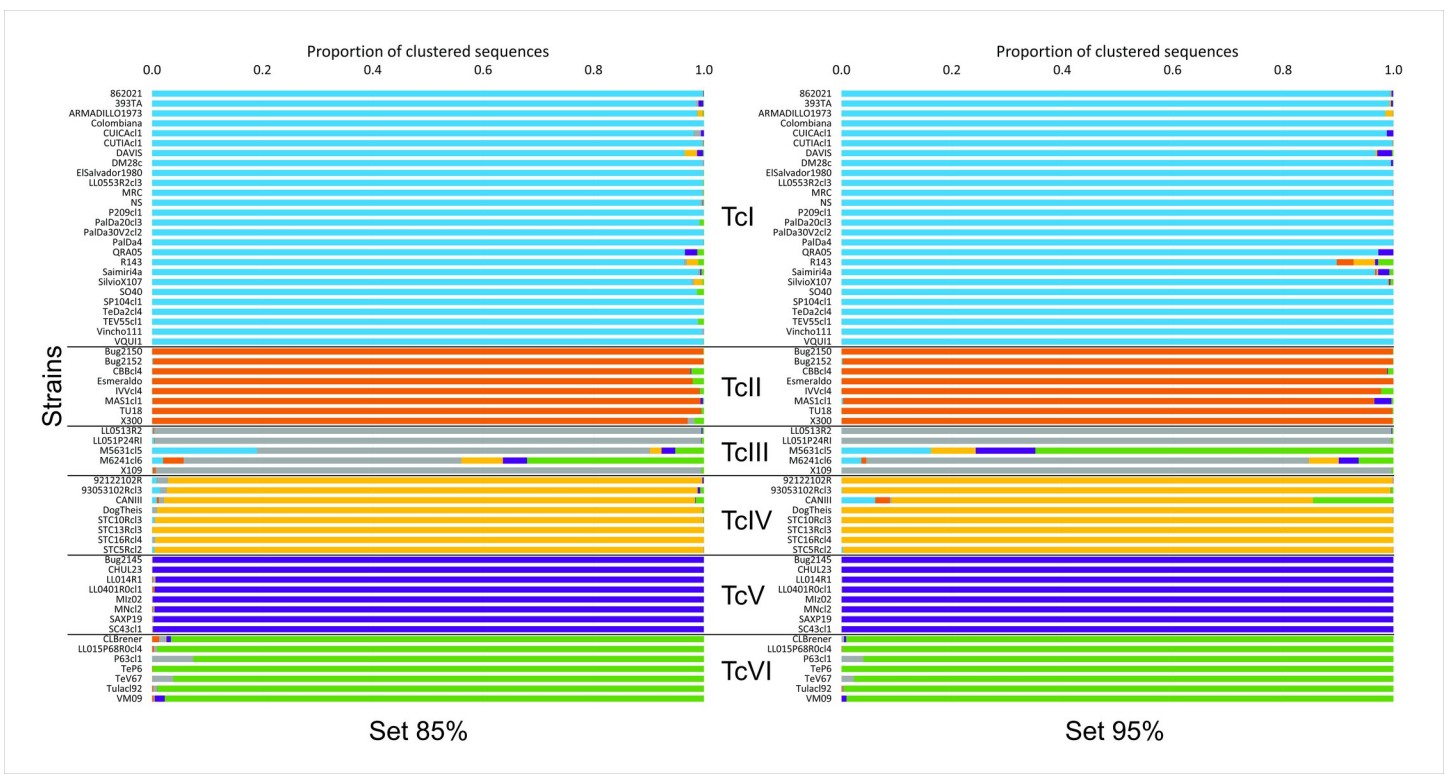

**Fig 2. The usefulness of sets of mHVR reference sequences for typing each strain.** The proportion of reads clustered with the reference sequences of each DTU is shown as horizontal bars for each strain. The color bars represent the proportion of reads that clustered with the reference sequences from each DTU. At the center, the DTU to which each strain belongs is indicated. Blue bars: TcI, orange bars: TcII, gray bars: TcIII, yellow bars: TcIV, violet bars: TcV, and green bars: TcVI. Each analyzed strain was typified using a reference set that excluded the sequences of the analyzed strain. Two different groups of reference sets were tested based on the mHVR clusters constructed with 85% (left) and 95% (right) similarity thresholds.

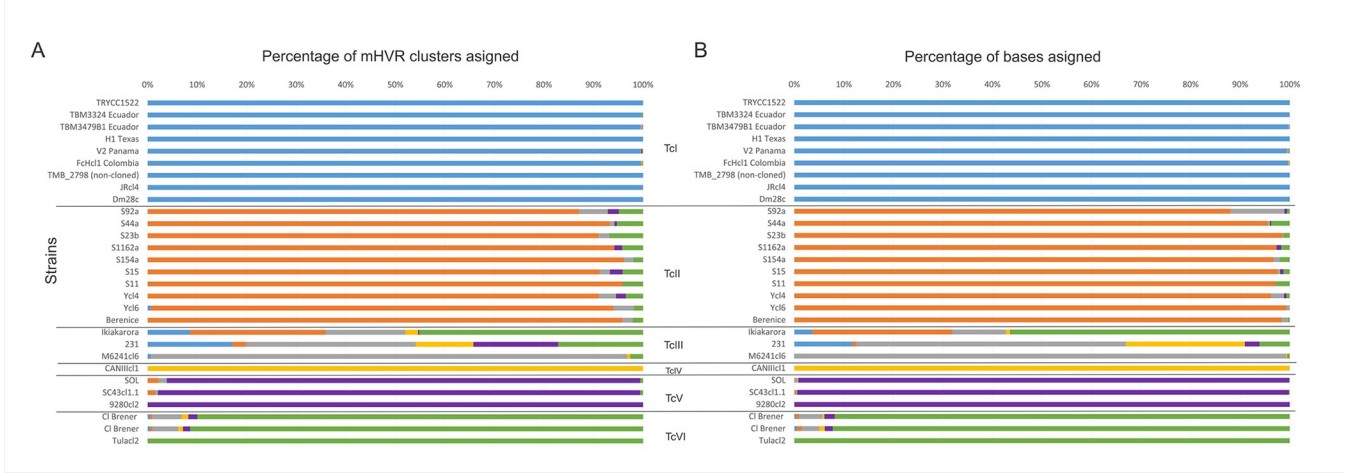

**Fig 3. The usefulness of the 95% mHVR reference sequences set for typing data from whole-genome projects.** The whole-genome reads for different strains were mapped to mHVR reference sequences of each DTU. A- The color bars for each strain represent the percentage of mHVR reference sequences for each DTU that were successfully mapped with a coverage of 270 bases at 10X depth. B- The color bars for each strain represents the percentage of the total number of bases for the whole-genome reads mapped to the mHVR reference sequences of each lineage with a coverage of 270 bases at 10X depth. At the center, the DTU to which each strain belongs is indicated. Blue bars: TcI, orange bars: TcII, gray bars: TcIII, yellow bars: TcIV, violet bars: TcV, and green bars: TcVI.

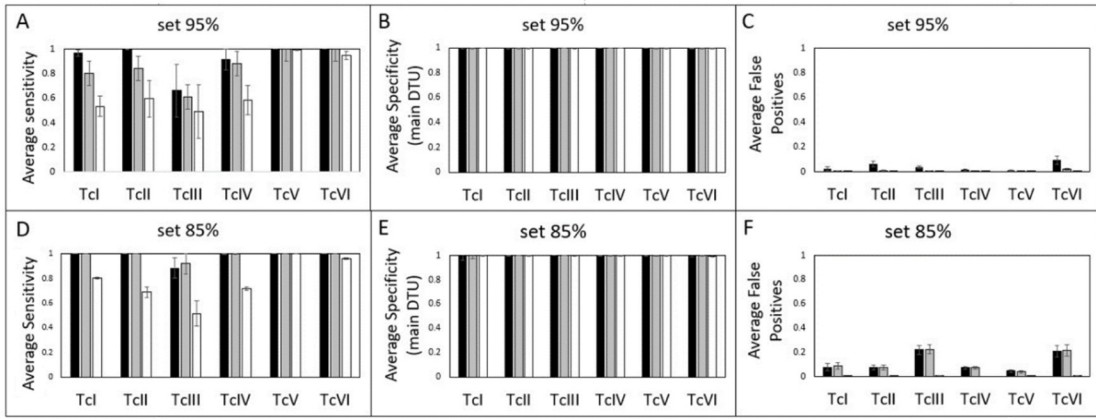

**Fig 4. The efficiency of reference sets for typing simulated PCRs.** A, D: Average sensitivity for the reference set constructed at the 95% similarity threshold for different simulated PCR conditions. B, E: Average specificity for typing DTUs based on the most abundant DTU-tag identified, while discarding minority DTU-tags in a sample for the 95% and 85% reference sets. C, F: Average false-positive rate for DTU detection considering all DTU-tags in a sample for the 95% and 85% reference sets. PCRs were simulated with 100 starting DNA molecules randomly selected from each strain dataset and 99% efficiency (black bars), 10 starting DNA molecules randomly selected from each strain and 80% efficiency (gray bars), and one randomly selected initial DNA molecule from each strain and 70% efficiency (white bars).

remained useful for typing when PCR had an efficiency of 80%-99% per cycle, starting with 10–100 molecules, with the 85% reference set producing better sensitivity results. Starting the PCR with a single molecule still allowed the typing of a sample, but with lower sensitivity but good specificity (Fig 4). Specificities were calculated considering only the most abundant DTU-tag in the sample. In other words, parasites from a certain DTU were considered present in a sample if their DTU-tag was the most abundant among the reads obtained from that sample. However, false positives were frequently observed for the secondary DTU in the sample. These false positives were more frequent when the 85% reference set was used (Fig 4).

Therefore, PCR simulations were used to define cutoff percentages for each secondary DTU identified based on the major DTU in the sample to reduce the risk of false positives for secondary DTUs in the sample (Tables 3 and 4). For example, for the 95% reference sequence set (Table 3) with a primary DTU-tag of TcI, the minimum frequency threshold to confirm the detection of a second DTU is 1% for TcII with an associated probability of error of 0.003.

**Table 3. Cutoffs for the minimum DTU-tag frequency indicate the presence of a secondary DTU in the sample according to the main DTU with the associated error probability for PCR simulations based on the 95% reference set.**

| | | Main DTU-tag | | | | | |
|---|---|---|---|---|---|---|---|
| | | **TcI** | **TcII** | **TcIII** | **TcIV** | **TcV** | **TcVI** |
| Secondary DTU-tags | TcI | | 0.01 (0.009) | 0.01 (0.01) | 0.02 (0.018) | 0.01 (0.003) | 0.01 (0.003) |
| | TcII | 0.01 (0.003) | | 0.01 (0) | 0.01 (0.003) | 0.01 (0.015) | 0.01 (0.017) |
| | TcIII | 0.01 (0.002) | 0.01 (0.001) | | 0.01 (0) | 0.01 (0.005) | 0.05 (0.06) |
| | TcIV | 0.01 (0.02) | 0.01 (0.001) | 0.01 (0.005) | | 0.01 (0) | 0.01 (0) |
| | TcV | 0.05 (0.017) | 0.03 (0.016) | 0.02 (0.013) | 0.01 (0.01) | | 0.03 (0.007) |
| | TcVI | 0.01 (0.009) | 0.05 (0.019) | 0.04 (0.02) | 0.03 (0.02) | 0.01 (0.005) | |

* Cutoff of the proportion of DTU-tags to reduce the error probability of misassigning a secondary DTU in the sample (error probability over 100 PCR simulations for each strain). Cutoffs were searched between 0.01–0.05 with 0.01 intervals. The maximum cutoff with an error probability is nearest to 0.02.

**Table 4. Cutoffs for the minimum DTU-tag frequency indicate the presence of a secondary DTU in the sample according to the main DTU with the associated error probability for PCR simulations based on the 85% reference set.**

| | | Main DTU-tag | | | | | |
| --- | --- | --- | --- | --- | --- | --- | --- |
| | | TcI | TcII | TcIII | TcIV | TcV | TcVI |
| Secondary DTU-tags | TcI | | 0,01 (0,01) | 0,05 (0,008) | 0,03 (0,016) | 0,01 (0,008) | 0.01 (0.007) |
| | TcII | 0,01 (0,003) | | 0,05 (0,043) | 0,01 (0,005) | 0.02 (0.009) | 0,05 (0,039) |
| | TcIII | 0,03 (0,01) | 0,03 (0,016) | | 0,03 (0,016) | 0.02 (0.006) | 0,05 (0,154) |
| | TcIV | 0,05 (0,023) | 0,01 (0) | 0,05 (0,088) | | 0,01 (0,004) | 0,01 (0,003) |
| | TcV | 0,04 (0,012) | 0,03 (0,015) | 0,05 (0,08) | 0,01 (0,018) | | 0,04 (0,011) |
| | TcVI | 0,03 (0,017) | 0,05 (0,044) | 0,05 (0,253) | 0,03 (0,015) | 0,01 (0,006) | |

These results show that the two reference sets have different utilities. The 85% reference set had better sensitivity for the majoritarian DTU in a sample, whereas the 95% reference set was less prone to false positives in the detection of co-infections.

## Detection of co-infections

A drawback of using cutoffs to reduce the risk of false-positive secondary DTU infection is the decrease in sensitivity for detecting such co-infections. For this reason, simulated mock samples built with different proportions of reads from different DTUs were evaluated to approximate the theoretical sensitivities for the detection of co-infections after applying the cutoff values. We analyzed the most common co-infections observed in patients, and the corresponding sensitivities are shown in Fig 5. The 85% and 95% reference sets had similar sensitivities for detecting secondary infections in a sample. However, some combinations of DTUs have shown very low sensitivity for the detection of co-infections. Consequently, the results suggest that the 95% reference set is preferable for detecting co-infection, with similar sensitivity to the 85% reference set but higher specificity.

## Usefulness on blood samples of infected patients

Building upon a prior study that examined the prevalence of different DTUs in blood samples from infected patients [34], we conducted deep amplicon sequencing of the mHVRs in such samples. The number of reads acquired for each of the twenty-eight samples, along with their corresponding DTUs, determined by using the 95% reference set, can be found in S3 File. These findings were compared with the Southern blot analysis using mHVR probes performed previously in such samples. Remarkably, amplicon sequencing identified at least one infecting DTU in all the samples (100%, 28/28), even in those with a low number of reads (Fig 6A and S3 File). Instead, the Southern blot method detected a DTU in 79% (22/28) of the samples (Fig 6A). Both techniques predominantly identified TcV as the most prevalent DTU, with frequencies of 27/28 and 22/28 for amplicon sequencing and Southern blotting, respectively (Fig 6B). A concordance rate of 82% was observed, and a Cohen's kappa index of 0.24 indicated a fair level of agreement between the methods. In addition, neither method detected the presence of TcII or TcIII in any sample. These results clearly showed that mHVR amplicon sequencing can be implemented in blood samples. Although TcI and TcVI were less prevalent, there was a noticeable discrepancy in their prevalence between the two techniques. Amplicon sequencing revealed a high prevalence of TcI compared to TcVI. For TcI detection, the concordance was 64% (kappa = 0.05), with most of the identifications attributed to amplicon sequencing (10 versus 2). Conversely, TcVI was detected more frequently using the Southern blot method than amplicon sequencing, with counts of 14 and 8, respectively. There was a notable

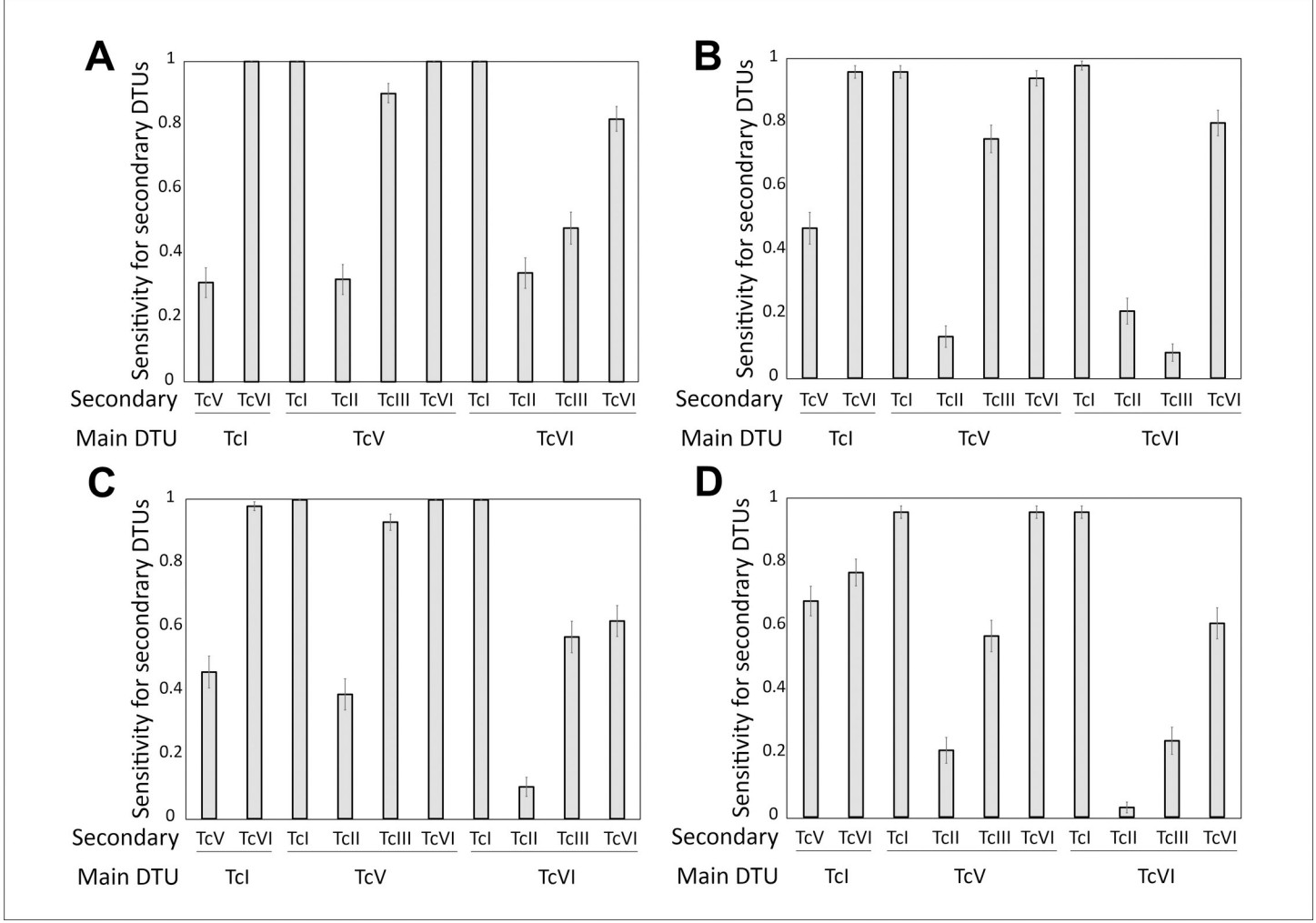

**Fig 5. Sensitivity for detecting secondary DTUs in the simulated samples.** A and B: Sensitivities using 95% reference set. C and D, sensitivities using an 85% reference set. Mock samples were simulated with different proportions of the main and secondary DTU. A and C: 90% of the reads of the main DTU and 10% of the secondary DTU. B and D: 95% of the reads of the main DTU and 5% of the secondary DTU. The strains used for each DTU were PalDa20cl3 (TcI), Esmeraldo (TcII), X109/2 (TcIII), MNcl2 (TcV) and LL015P68R0cl4 (TcVI).

discordance in the detection of TcVI between the two techniques (kappa = -0.15). As predicted, the 85% reference set was fully concordant with the detection of the main DTU in the sample when compared to the 95% reference set. However, a higher rate of secondary DTUs was also observed (S3 File).

## Discussion

Here, we present the development of a typing workflow based on deep amplicon sequencing of mHVRs amplicons from 62 strains belonging to the six main lineages of *T. cruzi*. The workflow allowed the use of two sets of mHVR reference sequences, one at 85% and another at a 95% similarity threshold, for different purposes. The 95% reference set has a higher specificity and is better suited for detecting co-infections. Instead, the 85% reference set is suitable for identifying the main DTU of a sample when the 95% reference set fails to detect a DTU. Firstly, we evaluated the workflow for its ability to genotype samples obtained from cultures and

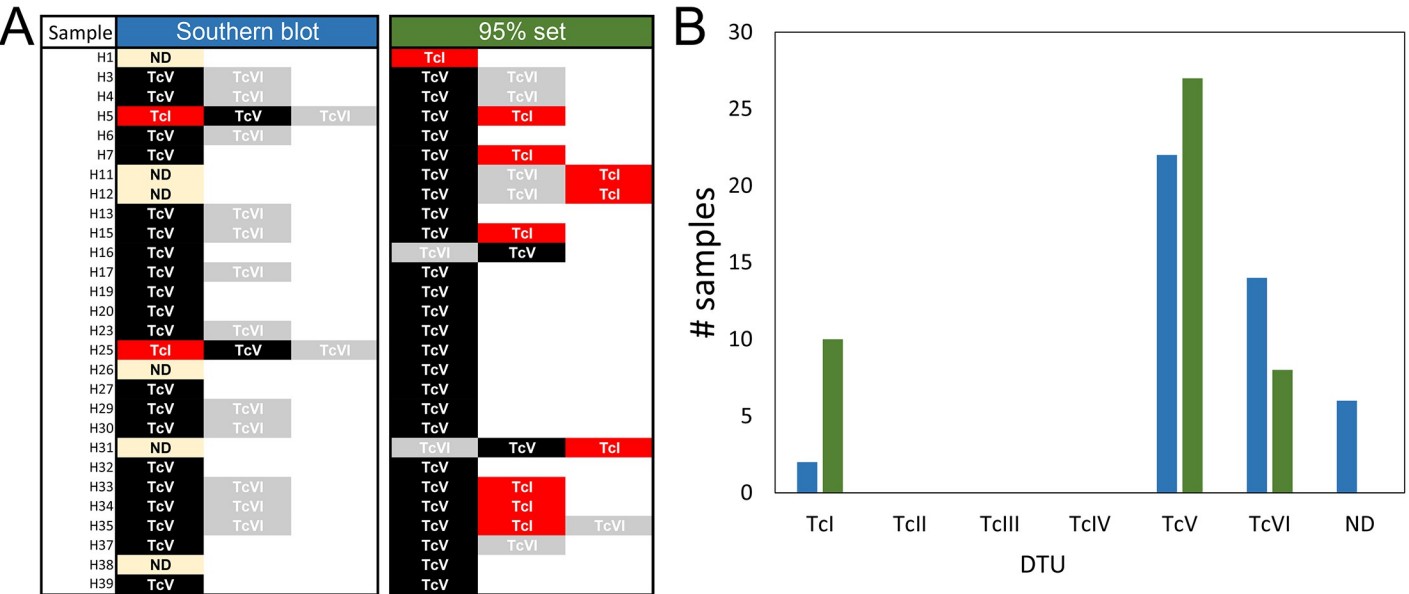

**Fig 6. Comparison between Southern blot and deep amplicon sequencing.** A, DTUs identified in blood samples of 28 patients by Southern blot by using mHVR probes (left) and deep amplicon sequencing by using the 95% reference set. B, comparison of the prevalence for different DTUs determined by using Southern blot (blue bars) and deep amplicon sequencing by using the 95% reference set (green bars). ND = nondetermined DTU by the method.

sequences sourced from the public domain such as genome data. Secondly, we assessed its performance under artificially simulated conditions, such as different PCR scenarios and in the presence of multiple infecting lineages. Finally, we addressed the workflow performance on blood samples derived from infected individuals.

To develop such a typing workflow, we preliminarily analyzed and compared the diversity of mHVRs in 62 reference strains of the six main DTUs. We observed that strains belonging to the same DTU shared most mHVR clusters, which is consistent with previous reports [42,43]. Moreover, most clusters were DTU-specific (Fig 1), indicating the potential use of these sequences for typing *T. cruzi* intraspecific diversity. Interestingly, the TcIV strains isolated in the USA shared most of their mHVR clusters, whereas CANIIIcl1 isolated in Brazil did not share mHVR clusters with the other TcIV strains (Fig 1). The difference can be attributed to the geographic origin of the analyzed strains, as it has been proposed that TcIV strains from North and South America have undergone phylogenetic divergence [44]. In contrast, TcI strains shared fewer mHVR clusters for the 95% similarity threshold, in contrast to TcV and TcVI strains, which shared most of the clusters (Fig 1B). In addition, we observed the geographic genetic structure among TcI strains, which is consistent with a previous study, in which a panel of samples with a wide geographic distribution was analyzed using a set of polymorphic microsatellite loci markers [11]. In addition, the 95% reference set also showed different mHVR composition patterns within some DTUs, which highlight their potential use at intra-DTU typing. Future studies encompassing a broader range of strains from diverse geographic regions and dedicated reference sets for each DTU will be required.

With the purpose of generating sets of reference sequences to be used for typing, we selected two sets of representative sequences based on the mHVR clusters generated for identity thresholds of 85% and 95%. The 85% mHVR reference sequence set was able to genotype all strains used in the analysis, possibly because of the lower sequence identity requirement. In addition, the 95% mHVR representative sequence set was able to genotype all the strains, with one exception. This set failed to type a TcIII strain (Fig 2), which could be attributed to several

factors. First, a 95% sequence set was generated from clusters built with a higher sequence identity threshold, resulting in more specific clusters. Additionally, the reference sequence set for TcIII lineage construction had a limited number of strains and thus may not be fully representative of the diversity within this lineage.

The 95% mHVR reference sequence set was evaluated for its suitability for typing sequences from genomic projects. The developed workflow allowed for typing of almost all analyzed strains, except for 231 and Ikiakarora strains -TcIII- (Fig 3). The percentage of mHVR reads from 231 strain that mapped to the TcIII lineage reference clusters was very low compared to another TcIII strain (M6241cl6). Although Ikiakarora (TcIII) exhibited high percentages of sequences mapping to the TcII and TcVI lineages, this appears to be a mixture of parasites from different DTUs or contamination. The results were promising, and a simple analysis could identify the lineage of a strain, co-infections, or patterns of genetic exchange. However, additional TcIII strains should be added to improve the sensitivity of the 95% reference set.

Our results further demonstrate that the 85% and 95% reference sets successfully and accurately typified the strains after a simulated PCR reaction which generates stochasticity on frequency of sequenced mHVRs. This was observed even under suboptimal simulated PCR conditions as low efficiency and few template DNA molecules. This suggests that both sets could be used for direct typing of biological samples with a low parasite burden, which is commonly observed in chronic patients; however, owing to the specificity the 95% reference set is preferable. Conventional multilocus PCR schemes [13–15] and parasite isolation only detect the most abundant DTU and overlook the diversity of parasites in the sample. Therefore, using mHVRs for typing could enable the detection of co-infections in patients, even if one of the infecting lineages is underrepresented in the sample. The high number of mHVR copies per parasite makes this approach more feasible for detecting co-infections. Our results also demonstrated that both sets of representative sequences were able to detect co-infections, although the 95% reference set was more specific in detecting a second DTU than the 85% reference set. However, for co-infections involving TcV/TcII, TcVI/TcII, and TcV/TcIII, a lower sensitivity was observed. In contrast to other approaches that are unable to differentiate between TcII-TcV-TcVI or TcV-TcVI [9,13,20,45], our method is able to accurately typified TcV and TcVI, even if they are presented as co-infecting DTUs.

We further assessed the amplicon sequencing efficacy using blood samples and found it to be proficient in assigning DTUs, even with a limited number of reads. When compared against Southern blotting using mHVR probes, our method showed overall good concordance. In particular, TcV detection had a high percentage concordance (82%), although with a low kappa index (0.239). It is important to consider that Cohen's kappa accounts for chance agreement, meaning it adjusts for the agreement that would be expected just by chance. However, this index was influenced by the prevalence of DTU infection. If the prevalence of a DTU is either very high (as observed for TcV) or very low, chance agreement is also high and kappa is reduced accordingly [46]. This explains the high agreement percentage and low kappa index for TcV detection. In addition, certain discrepancies were noted, particularly regarding the secondary DTUs present in the samples. Amplicon sequencing identified a higher prevalence of TcI than Southern blotting. This was expected because the TcI mHVR probe for Southern blotting was built with mHVR amplicons of a unique TcI strain [34]. Instead, amplicon sequencing was based on sequences from 26 TcI strains, which enhanced its sensitivity. Conversely, TcVI was detected more frequently by Southern blotting than by our amplicon sequencing method. This result was unexpected because TcVI has a relatively low genetic diversity, and consequently, it would be expected to be fairly represented in the 95% reference set. It is important to note that the specificity of the Southern blot method was evaluated with a reduced dataset (less than 20 strains) [34] and that hybridization can be sensitive to probe

incubation conditions. Consequently, the discordance may be attributed to the cross-reaction of the mHVR TcVI probe in the Southern blot.

A potential drawback of our method is related to minicircle inheritance. Maxicircles from *T. cruzi* kDNA have been suggested to be inherited uniparentally, whereas we previously proposed that minicircles are inherited biparentally in hybrids [21,47]. If minicircles are inherited biparentally, they are expected to behave similarly to the nuclear genes. Therefore, we anticipate that the typing results will be similar to those obtained using other methods that use nuclear markers. However, when maxicircle markers are used for typing, divergent outcomes are anticipated in cases of hybridization or mitochondrial introgression. In this sense, it is crucial not to overlook the numerous instances of mitochondrial introgression that have been reported [48].

Overall, our findings suggest that using mHVRs for the direct typing of clinical samples could be a promising strategy for identifying and characterizing co-infections caused by different *T. cruzi* lineages. Our simulation-based approach provides valuable insights and demonstrates the potential for coinfection detection. However, further experimental validation is required to ensure reliability and applicability for detecting coinfections. Laboratory experiments using artificial samples created by mixing DNA from different known strains would be necessary.

Numerous typing assays have been proposed for *T. cruzi*. However, there is an increasing need for simpler and more cost-effective methods. In this regard, the proposed typing workflow based on mHVR amplicon sequencing is suitable for simultaneously typing hundreds of samples and based on sequences, unlike other techniques that use the same target. Furthermore, this typing workflow is more economical than other techniques [11,16,17,19,20,49,50] and the bioinformatics analysis is relatively simple because it only needs to upload the data on a Google colaboratory notebook (no specific hardware and no bioinformatic skills are required) and follow the steps until the typing results are obtained. Moreover, the workflow would be appropriate for direct biological typing because only one PCR reaction is required to generate the libraries; in contrast to typing schemes, it would also greatly contribute to answering questions related to the clinical manifestations of Chagas disease. However, issues related to the sensitivity of detecting certain DTUs in co-infections still need to be resolved. Overcoming this lack of sensitivity can be achieved by adding new strains to the reference sets, particularly strains with underrepresented DTUs.

In conclusion, the proposed workflow offers a simple, low-cost, and efficient alternative for typing *T. cruzi* strains, and its potential applications in clinical and epidemiological studies are promising. Finally, future applications of deep sequencing of mHVR amplicons will help refine the workflow and clarify its limitations and impact areas.

## Supporting information

**S1 Table. Reads obtained after different steps in the pipeline.**
(PDF)

**S2 Table. True and false-positive rates for different reference sets on reads and strains.**
(DOCX)

**S1 Fig. The usefulness of the 95% set of mHVR reference sequences for typing data from whole-genome projects.** The whole-genome reads for different strains were mapped to mHVR reference sequences of each DTU. A- The color bars for seach strain represent the percentage of mHVR reference sequences for each DTU that were successfully mapped with a coverage of 170 bases at 10X depth. B- The color bars for each strain represents the percentage

of the total number of bases for the whole-genome reads mapped to the mHVR reference sequences of each lineage with a coverage of 170 bases at 10X depth. At the center, the DTU to which each strain belongs is indicated. Blue bars: TcI, orange bars: TcII, gray bars: TcIII, yellow bars: TcIV, violet bars: TcV, and green bars: TcVI.
(JPG)

**S1 File. Evaluation of the PCR simulating algorithm by comparison against a duplicate experimental PCR of mHVRs from the LL015P68R0cl4 strain (TcVI).**
(PDF)

**S2 File. Proportion of mHVR clusters shared between different genomes and different strains in the 95% reference dataset.**
(XLSX)

**S3 File. Reads obtained from blood samples and percentages of clustering against each DTU.**
(XLSX)

## Author Contributions

**Conceptualization:** Patricio Diosque, Nicolás Tomasini.

**Data curation:** Fanny Rusman, Anahí G. Díaz, Tatiana Ponce, Noelia Floridia-Yapur, Christian Barnabé.

**Formal analysis:** Fanny Rusman, Anahí G. Díaz, Nicolás Tomasini.

**Funding acquisition:** Patricio Diosque, Nicolás Tomasini.

**Investigation:** Fanny Rusman, Anahí G. Díaz.

**Methodology:** Fanny Rusman, Anahí G. Díaz, Nicolás Tomasini.

**Resources:** Patricio Diosque.

**Supervision:** Nicolás Tomasini.

**Writing – original draft:** Fanny Rusman.

**Writing – review & editing:** Anahí G. Díaz, Tatiana Ponce, Noelia Floridia-Yapur, Christian Barnabé, Patricio Diosque, Nicolás Tomasini.

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
