## [Decision Letter · Decision Letter 0]

5 Jul 2023

Dear Dr. Tomasini,

Thank you very much for submitting your manuscript "Wide reference databases for typing Trypanosoma cruzi based on amplicon sequencing of the minicircle hypervariable region" for consideration at PLOS Neglected Tropical Diseases. As with all papers reviewed by the journal, your manuscript was reviewed by members of the editorial board and by several independent reviewers. In light of the reviews (below this email), we would like to invite the resubmission of a significantly-revised version that takes into account the reviewers' comments. 

As pointed out by the reviewers, the major discrepancies in the scope of the manuscript need to be addressed, from the generation of a database suggested in the title, but not presented and rather proposed as a follow-up work later in the discussion, to the claim of a new typing method/strategy, but no experimental validation with biological samples is presented. The additional comments from the reviewers should also help improve the study.

We cannot make any decision about publication until we have seen the revised manuscript and your response to the reviewers' comments. Your revised manuscript is also likely to be sent to reviewers for further evaluation.

Sincerely,

Eric Dumonteil, Ph.D.

Academic Editor

Paul Brindley

Editor-in-Chief

As pointed out by the reviewers, the major discrepancies in the scope of the manuscript need to be addressed, from the generation of a database suggested in the title, but not presented and rather proposed as a follow-up work later in the discussion, to the claim of a new typing method/strategy, but no experimental validation with biological samples is presented. The additional comments from the reviewers should also help improve the study.

Reviewer's Responses to Questions

**Key Review Criteria Required for Acceptance?**

**Methods**

-Are the objectives of the study clearly articulated with a clear testable hypothesis stated?

-Is the study design appropriate to address the stated objectives?

-Is the population clearly described and appropriate for the hypothesis being tested?

-Is the sample size sufficient to ensure adequate power to address the hypothesis being tested?

-Were correct statistical analysis used to support conclusions?

-Are there concerns about ethical or regulatory requirements being met?

Reviewer #1: The objectives and hypothesis are clearly stated in the manuscript and are both evaluated through an appropriate experimental design and strain population.

While the species Trypanosoma cruzi is highly diverse, the molecular typing methods studied so far have not been sufficient to describe the intra-DTU variability, thus hindering the evaluation of the association of genetic variants with clinical-epidemiological variables.

The methodology is detailed both in the population description. The wet lab methodology and particularly the data analysis is clearly described and elaborated.

Reviewer #2: 0. Abstract

Do not use abbreviations in the abstract (mHVR) or if at all necessary please use the complete description first, e.g. "minicircle hypervariable region (mHVR)". 

1. Not serious. But still major. The manuscript title starts with "Wide reference databases..." and then the abstract follows with "We present reference databases of mHVR sequences ..." however there is **apparently** no data or database accompanying the manuscript, nor a file deposited anywhere (data dryad, zenodo, figshare). 

However, it seems that the data and code is available only in the github repository mentioned in methods (lines 138-139). It is a little hidden gem in the Ms. There is also a nice python notebook. Maybe the authors like to increase the visibility of these resources? I believe these are the major outputs, and would increase the impact of the paper and provide a fast route for readers so they can go from sequencing to typing. It is in the interest of authors to make this as easy to use and apply as possible! 

Maybe add a webpage (in github?) with a tutorial on how to use the notebook and these databases? Provide these in the Ms in a more prominent section (DATA AND CODE AVAILABILITY or maybe better REFERENCE DATASETS AND CODE). In any case, you get the idea, currenly the fact that the database is inside the github repo gets lost in the methods, and also the github repo is only mentioned at the end of the "Using the reference datsets".

2. PCR simulation

Methods, page 14, lines 175-203

Is there any reference for the algorithm simulating the PCR? What is the value of e in formula (2) (line 188). In the description the value of m(i) is obvious, as well as the value of n(i-1). However the value of e is described as "the duplication probability of each DNA molecule determined according to the PCR efficiency", but why did you choose the values in line 203 for the PCR efficiency (0.7, 0.8, 0.99)? Please clarify. Is there any published paper describing this type of simulation? Why these values? Please provide references. 

Have this algorithm been validated experimentally by the authors and/or by others?

3. Results - Threshold similarity percentages

Why were these two values chosen? (85%, 95%). What was the rationale for these? In the manuscript there are only vague explanations. 

Methods, (line 109) "Then, sequences were clustered at 85% and 95% pairwise identity..." Why these thresholds?

Results, (lines 228-229) "The reads were clustered according to sequence similarities using different minimum similarity percentages (85% and 95%). [...] number of shared clusters among strains decreased when higher similarity thresholds were used. For the 85% similarity threshold, it was observed that TcI lineage strains, which were geographically closer, shared more mHVR clusters than strains isolated at greater geographical distances. Additionally, at the 95% similarity threshold, most TcI strains shared a few mHVR clusters. In contrast, the TcV and TcVI strains still shared mHVR clusters with other strains of the same DTU. These results suggest that mHVR sequences can be used for typing T. cruzi strains." 

Have you considered running the analysis over a range of similarity thresholds (e.g. 70% to 99% in steps of 1%, 2%, 5% don't know what would be a sensible step size but you get the idea), and measure some metric (sensitivity/specificity) to find the best threshold? Maybe the Area Under the ROC curve (AUC)? Maybe the authors already did this? In this case please provide these data (supplementary) as support for these chosen thresholds.

Results, page 18, 95% reference set for typing strains from whole-genome sequencing data. (lines 264-276). 

Why was this threshold used? I'm trying to make sense on how to move from "Reference sets constructed with a 95% similarity threshold failed to typify one TcIII strain (M5631cl5)." (lines 253-254, previous section) to "To address the suitability of the 95% mHVRs reference set for typing..." (lines 264-265, beginning of next section). Maybe put out loud and clear what is the rationale behind this? I am a little bit confused because although a stricter 95% reference set is used (perhaps aiming to match at the sub-DTU level?), Figure 3 (and Supp Fig 1A) only show mapping (typification) of strain data (from genome sequencing projects) onto DTU-level groups (same as done with the 85% reference set). 

Maybe I'm missing something here? I was perhaps expecting a number of figures (supplementary of course) similar to Fig3 but where 1) query genome is one strain/isolate per figure; and 2) the Figure shows both panels A + B (percentage reads, percentage of bases) for _all_ strains and isolates that make up the 95% reference set. In an hypothetical case of e.g. using CL-Brener (IPB/CSIC, IonTorrent, see below) as query, this figure would show the percentage of reads and bases matching each of the strains in the reference dataset. From here one should hopefully be able to see that CL-Brener matches itself and also maybe the other CL-Brener instance (UFMG/Brazil, Illumina) first (top-ranked) and then down maybe another TcVI strain, etc. This type of figure should show sub-DTU level matching of individual strains (if I understood clearly the aim of the 95% reference set)

4. Genome data used

Table 2, page 12, lines 146-148.

Why were these genomes selected? Was there any rationale to omit other genomes? This seems strange because, there are a number of Sylvio X10 genomes (DTU TcI) sequenced by different groups. Similarly for Dm28c (DTU TcI), and TCC (a CL-derivative strain), amongst others. Having the same strain/isolate sequenced independently would provide robust validation of the methodology and the developed reference datasets. Please clarify. 

Also here, what is the difference between the two CL-Brener genomes and why did you include these two? I see one is from the IPB/CSIC (Spain), sequenced using IonTorrent. And the other is from UFMG (Brazil), sequenced using Illumina. Besides clarifying why the authors included two datasets from the same strain, maybe the authors would like to expand on analyzing the ability to typify strains using either IonTorrent vs Illumina data?

Reviewer #3: The rationale of the study is appropriate, the high sequence diversity and copy number of the high variabLe region of the minicircles of T.cruzi ( mHVR ) makes it a good marker for parasite genotyping at the DTU and infra_DTU levels. These aims are partially addressed by the experimental work because analyses has been done with DNA from reference strains, with datasets from T.cruzi genome projects and with mock samples simulating mixed DTUs. The authors have found adequate typing results with some limitations, in particular for the low number of strains for a given DTU such as Tc III. So, the sample size should be increased in particular for this DTU. 

Statistical analyses are clearly described . 

Which was the criterion to select 85% and 95% values as minimum similarity percentages instead of looking for the optimal minimum percentage to have two categories of classification : one that englobes all strains from a same DTU in a single cluster and accurately separate DTUs, and another one that could be used to distinguish subDTUs in a given population ? 

In my opinion, the work would significantly enrich if this typing algorithm is applied to a panel of true biological samples already characterised using a previously reported method, to be able to detect degree of agreement and discordances.

**Results**

-Does the analysis presented match the analysis plan?

-Are the results clearly and completely presented?

-Are the figures (Tables, Images) of sufficient quality for clarity?

Reviewer #1: The results are clear and organized according to the aim of the study. Images and tables are correctly visualized and clear.

Reviewer #2: -Does the analysis presented match the analysis plan?

YES

-Are the results clearly and completely presented?

YES

-Are the figures (Tables, Images) of sufficient quality for clarity?

Figure 2. Font / letter size of axes labels (e.g. "Strains"; "Set 85%", "TcI", "TcII") is disproportionately big in comparison with the tiny size of text (unreadable at default zoom level) containing strain / isolate names (e.g. "862021", "Armadillo1975", "Colombiana")

Figures 3 & Supplementary Figure 1. Are these two the same? What is the difference between these two?

Reviewer #3: By comparing the diversity of mHVRs in 62 reference strains of the six main DTUs, at 80,000 reads of depth, the authors confirmed observations of previous works that employed other markers, as it is mentioned by them in the discussion section. 

The 95% mHVR reference sequence set was evaluated for its suitability for typing sequences from genomic projects with good results. 

Authors demonstrated that the representative sequence sets at 85% and 95% accuracy were both able to identify mock samples. It would be nice to add at least a panel of biological samples to be able to characterize better the limitations of the technique in the presence of host DNA of different procedences and in real mixed infections, for example with triatomines coinfected with different DTUs. For now, the usefulness of the 95% reference set in such an application is an hypothesis.

As DTU 1 is so variable and distributed into different clusters, could be proposed from these data, together with data from others ( for example studies using microsatellites) , that DTU I actually might be conformed by more than one DTU ( eg. Ia, Ib...I n?) while DTU V and VI, for example are much more homogeneous ? 

Figures

Labels of strains in Figures 2 and 3 need better resolution to be clearly read

**Conclusions**

-Are the conclusions supported by the data presented?

-Are the limitations of analysis clearly described?

-Do the authors discuss how these data can be helpful to advance our understanding of the topic under study?

-Is public health relevance addressed?

Reviewer #1: The conclusions are in accordance with the results and limitations are clearly expressed by the authors. These limitations are presented as future objectives of the group.

The scope of the study was not only the generation of new data about T. cruzi but also build a database to enable the study of this strategy worldwide.

Reviewer #2: 5. Discussion

page 23, lines 357-360

"Moreover, the workflow was evaluated for its ability to typing samples derived from cultures, a set of complete genome data, under different simulated PCR conditions, and in the presence of more than one infecting lineage."

I don't like this sentence. I understand what the authors are trying to state, but I would split this sentence in two: one sentence for the cases where you've used experimentally determined sequences (sequencing samples from cultured strains; sequences obtained from the public domain); and one for the artificially simulated samples (simulated PCR conditions; simulated co-infections). This way it should be more clear to readers.

Also I don't quite like how the PCR-results are discussed. Not trying to come hard on authors, you do say "simulated samples" often. So it should be clear. But I'd prefer to read "simulated PCR reactions" or "simulated PCR experiments" as this come closer to what the authors did, which is to 1) generate artificial DNA samples (templates) for PCR; and 2) run a simulated PCR amplification in the computer. 

I would also like to see a statement saying that this of course should be validated experimentally (detecting co-infections), maybe even using artificial (spiked) samples created in the lab (not in the computer) by mixing DNA from different strains. I am not asking that the authors do the real PCR validation for this Ms (great if you do, but not required for acceptance), only that you mention this fact.

6. Further Discussion

There are two additional issues I don't see discussed here. 

mHVR amplicons are derived from mitochondrial DNA (kDNA). And inheritance of mitochondrial DNA may not follow nuclear DNA. Because other schemes and methods for typification of DTU are based on nuclear-DNA markers, I think it merits some discussion here on whether this is something that should be paid attention to, or not (and why?) I see the authors have some published analysis on this (ref #21). A brief discussion here would help readers put this in context. 

Amplicon sequencing is OK for cultured strains. And simulated PCR experiments are OK in this Ms to show something more towards application in a real world setting (clinical samples from patients). But this is not discussed much, and I think this is one of the major impacts perhaps in the long term. Maybe the authors would like to discuss something about performing PCR amplification of mHVR regions from clinical samples? Performance? Also maybe discussing this in the context of typical focal infections produced by T. cruzi where different infection foci in the body may activate at different times (e.g. see PMID:32799361). Is there something the authors could say on the applicability of this method to clinical samples? Future prospects?

Reviewer #3: Conclusions are supported as a feasibility study of the use of this typing algorithm but further studies using panels of biological samples and blind evaluation in the field will be necessary to demonstrate the potential of this proposal for epidemiological studies. A more robust work will be presented when the researchers will expand the reference sets of mHVRs to some DTUs and report the online database for automated data analysis.

**Editorial and Data Presentation Modifications?**

Reviewer #1: (No Response)

Reviewer #2: MINOR ISSUES / COMMENTS

Introduction, page 7, lines 69-70

"Due to the low level of parasites circulating in the peripheral blood or infected tissues in chronically infected patients, most typing methods have limited sensitivity (24)."

ref #24 does not describe typing methods or performance at the task of discriminating DTUs, it only describes performance of PCR for _detection_ of T. cruzi using 6 DNA targets: kDNA, Sat-DNA, 24S, CO-II, SL-DNA, 18S. 

Methods, page 9, lines 99-100 

"A Fragment Analyzer (Advanced Analytical Technologies, USA) was used to validate the libraries."

I have been unable to find this company. Please clarify which fragment analyzer was used in the study (model? Part number? manufacturer?). Alternatively describe the quality metrics used to assess nucleic acid integrity. 

Methods, page 11, lines 111-112 and 115-116

"outputs (seqs_otus.txt and the otu table) were filtered using filter_otus_from_otu_table.py"; "The most abundant sequence in each mHVR cluster was selected as the representative sequence using the “pick_rep_set.py” script"

Are both scripts available as part of the QIIME software or just the first one? Please clarify. 

Also here, lines 114-118, the text is confusing: 

"The most abundant sequence in each mHVR cluster was selected as the representative sequence" (this implies there is only one representative sequence per cluster), but then the following sentence: "The output contained a representative set of sequences for each mHVR cluster..." is saying that there is _a set of sequences_ (several) for each mHVR cluster. Please clarify. 

Methods, page 13, lines 154-155

"The reads were mapped against the reference set of mHVR at 95% similarity using BWA-MEM (36) with default parameters."

Please report the version of bwa-mem. Otherwise the phrase "with default parameters" may make no sense. Alternatively, please report and define these default parameters.

Methods page 14, lines 176-177

"First, the algorithm random samples s molecules from a multinomial distribution"

Maybe rephrase to "First, the algorithm samples s random molecules from a multinomial distribution"?

Results, page 17, lines 224-225

To address the suitability of the analysis of mHVR sequences in a sample infected with T. cruzi, it was first evaluated [which?] mHVR sequences [were?] shared between different cultured strains.

Results, page 19, line 274

"Notably, all evaluated strains were accurately typing [typified?] using this approach"

Results, page 19, line 297

"a DTU would be infecting a sample" infecting? what is the idea behind a DTU "infecting" a sample? Please clarify or use another word to convey your idea. 

Discussion page 25, lines 377, 379

"The 85% mHVR reference sequence set was able to typing all strains" 

"the 95% mHVR representative sequence set was unable to typing one TcIII strain"

typify all strains? typify one TcIII strain?

Reviewer #3: ACCEPT

**Summary and General Comments**

Reviewer #1: I only have one concern about the typing method proposed by this study, and this is the cost of NGS in endemic areas of Chagas disease. Perhaps the authors can propose/think of a method based on the obtained data that is applicable in resource-limited contexts, which happen to coincide with the endemic regions of the disease.

Reviewer #2: The work by Rusman F et al is a nice and timely description of both methods and a reference data set that would allow other labs to perform amplicon sequencing of T. cruzi DNA samples to perform DTU assignment and/or detect co-infections. 

The work is original, well-written, and provides a well-described method and code to perform strain and DTU-typing on DNA samples. The authors have validated their method and their reference databases by typing available genomes in the public domain as well as by performing simulated PCR experiments in the computer with different levels of sub-sampling of the original (template) DNA. 

That said, I do have some comments and suggestions (see below). My recommendation is acceptance after revision. Congratulations to the authors on a clean and nice to read Ms!

Reviewer #3: This is a well written interesting approach to improve genotyping of T.cruzi based on deep sequencing of the highly variable region of the multicopy minicircle DNA of T.cruzi , showing promising resolution when working with DNA from reference strains, data from genome projects and mock samples. Further studies using panels of biological samples and blind evaluation of this strategy in the field is still needed to demonstrate its applicability in epidemiological studies. A more robust work will be presented when the researchers will expand the reference sets of mHVRs to some DTUs and report the online database for automated data analysis.

PLOS authors have the option to publish the peer review history of their article (what does this mean?). If published, this will include your full peer review and any attached files.

Reviewer #1: No

Reviewer #2: Yes: Fernan Aguero

Reviewer #3: No
---

## [Decision Letter · Decision Letter 1]

17 Oct 2023

Dear Dr. Tomasini,

Thank you very much for submitting your manuscript "Wide reference databases for typing Trypanosoma cruzi based on amplicon sequencing of the minicircle hypervariable region" for consideration at PLOS Neglected Tropical Diseases. As with all papers reviewed by the journal, your manuscript was reviewed by members of the editorial board and by several independent reviewers. The reviewers appreciated the attention to an important topic. Based on the reviews, we are likely to accept this manuscript for publication, providing that you modify the manuscript according to the review recommendations. 

Sincerely,

Eric Dumonteil, Ph.D.

Academic Editor

Paul Brindley

Editor-in-Chief

Reviewer's Responses to Questions

**Key Review Criteria Required for Acceptance?**

**Methods**

-Are the objectives of the study clearly articulated with a clear testable hypothesis stated?

-Is the study design appropriate to address the stated objectives?

-Is the population clearly described and appropriate for the hypothesis being tested?

-Is the sample size sufficient to ensure adequate power to address the hypothesis being tested?

-Were correct statistical analysis used to support conclusions?

-Are there concerns about ethical or regulatory requirements being met?

Reviewer #1: (No Response)

Reviewer #2: Accept the responses provided, and the revised manuscript with the changes.

**Results**

-Does the analysis presented match the analysis plan?

-Are the results clearly and completely presented?

-Are the figures (Tables, Images) of sufficient quality for clarity?

Reviewer #1: (No Response)

Reviewer #2: Accept the responses provided, and the revised manuscript with the changes.

**Conclusions**

-Are the conclusions supported by the data presented?

-Are the limitations of analysis clearly described?

-Do the authors discuss how these data can be helpful to advance our understanding of the topic under study?

-Is public health relevance addressed?

Reviewer #1: (No Response)

Reviewer #2: Accept the responses provided, and the revised manuscript with the changes.

**Editorial and Data Presentation Modifications?**

Reviewer #1: (No Response)

Reviewer #2: Regarding the new Supp File 1 and the additions to Methods and Results, I thank the authors for including these additional data and experiments. However I had trouble reading and figuring this all out, so I have one final suggestion (which the authors may ignore), which is to try and clarify this a little bit more for the readership. Below are my notes: 

line 252, methods 

"PCR model was compared to empirical data of mHVR cluster abundances for two independent PCR reactions of the same sample (Supplementary File 1)."

maybe substitute "empirical" with "experimental"?

line 566. Supporting information (legend to figures?) "Supplementary File 1. Evaluation of the PCR simulating algorithm by comparison against PCR repetition of the strain LL015P68R0cl4."

The text in the response to us reviewers is more clear in explaining this, maybe add some more detail, e.g. as in 

"Supplementary File 1. Evaluation of the PCR simulating algorithm by comparison against a duplicate experimental PCR of mHVRs from the LL015P68R0cl4 strain (TcVI)." Also, the legend in Supp File 1 says the strain is LL015P68R0 (add cl4?)

Supplementary File 1. What is the x axis in the plots? PCR cycles? What is the scale? There are no labels or tick marks for this axis. This should be clarified and added.

lines 424-425, 430, 434. There is something wrong with the kappa index here. The kappa index is used many times but it is not defined anywhere. Is this Cohen's Kappa? If so, it is confusing that there is a concordance rate of 82% with a kappa index of 0.24 (which suggests low agreement between the two techniques). Is there something wrong here? or this at least merits some discussion. Maybe it is just that percent agreements are not robust in comparison with cohen's kappa?

**Summary and General Comments**

Reviewer #1: The authors have addressed all the questions and suggestions from the reviewers and have made the relevant modifications to the manuscript

Reviewer #2: Accept the responses provided, and the revised manuscript with the changes.

PLOS authors have the option to publish the peer review history of their article (what does this mean?). If published, this will include your full peer review and any attached files.

Reviewer #1: No

Reviewer #2: Yes: Fernán Agüero

Figure Files:

Data Requirements:

Reproducibility:

References

---

## [Editor Report · Decision Letter 2]

2 Nov 2023

Dear Dr. Tomasini,

We are pleased to inform you that your manuscript 'Wide reference databases for typing Trypanosoma cruzi based on amplicon sequencing of the minicircle hypervariable region' has been provisionally accepted for publication in PLOS Neglected Tropical Diseases.

Best regards,

Eric Dumonteil, Ph.D.

Academic Editor

Paul Brindley, Ph.D.

Editor-in-Chief

---

## [Editor Report · Acceptance letter]

8 Nov 2023

Dear Dr. Tomasini,

We are delighted to inform you that your manuscript, "Wide reference databases for typing Trypanosoma cruzi based on amplicon sequencing of the minicircle hypervariable region," has been formally accepted for publication in PLOS Neglected Tropical Diseases.

Best regards,

Shaden Kamhawi

co-Editor-in-Chief

Paul Brindley

co-Editor-in-Chief
